# DEEP EXPECTATION-MAXIMIZATION IN HIDDEN MARKOV MODELS VIA SIMULTANEOUS PERTURBATION STOCHASTIC APPROXIMATION

## ABSTRACT

We propose a novel method to estimate the parameters of a collection of Hidden Markov Models (HMM), each of which corresponds to a set of known features. The observation sequence of an individual HMM is noisy and/or insufficient, making parameter estimation solely based on its corresponding observation sequence a challenging problem. The key idea is to combine the classical Expectation-Maximization (EM) algorithm with a neural network, while these two are jointly trained in an end-to-end fashion, mapping the HMM features to its parameters and effectively fusing the information across different HMMs. In order to address the numerical difficulty in computing the gradient of the EM iteration, simultaneous perturbation stochastic approximation (SPSA) is employed to approximate the gradient. We also provide a rigorous proof that the approximated gradient due to SPSA converges to the true gradient almost surely. The efficacy of the proposed method is demonstrated on synthetic data as well as a real-world e-Commerce dataset.

## 1 INTRODUCTION

Hidden Markov Model (HMM) is a powerful statistical framework in sequential data analysis. One advantage of HMM lies in its strong model interpretability (Rabiner, 1989). Estimating the parameters of a HMM given its observation sequence is a well-studied problem. The celebrated Expectation-Maximization (EM) algorithm (Dempster et al., 1977) finds a locally optimal estimation in the maximum likelihood sense with provable optimality (Wu, 1983). Numerous modifications and enhancements to the EM algorithm have been proposed (Jamshidian & Jennrich, 1997; Capp, 2011). Notably, explicitly modeling the duration of the states in a HMM can be enabled by an extension named Hidden Semi-Markov Models (HSMM) (Ostendorf et al., 1996).

This paper concerns estimating the parameters of a collection of HMMs, each of which corresponds to a set of known features. The observation sequence of each individual HMM is noisy and/or insufficient, making parameter estimation solely based on its corresponding observation sequence using the traditional EM algorithm impractical. The problem arises in recommender systems of online E-Commerce platforms. The objective is to train a HMM (HSMM) model to predict the category of product that the user would purchase based on the user's feature (demographic, location, income level) and observation sequence (past click history). The observation sequence of a particular user is generally noisy (click not reflective of true purchase intention) and insufficient (user only clicked a small number of items), making parameter estimation highly challenging.

The intuition is that while the information regarding each individual user is quite limited, we have access to a much larger amount of information concerning all the users collectively. If we were able to quantify some kind of correlation among the users using their features, we can combine their information for parameter estimation. Formally, we propose to deduce correlations among the HMMs using their features, so that we can fuse the information across different HMMs to estimate the parameters. The basic idea is to combine the classical Expectation-Maximization (EM) algorithm with a neural network which maps a HMM's feature to its parameters. The weight of the neural network captures the correlation among the HMMs and is shared across all the users. The EM iteration, with the estimated HMM parameter at step $k$ (denoted by $\lambda_k$) as input and the refined

parameter $\lambda_{k+1}$ as output, can be considered as a building block (or layer) in the resulted neural network. In order to enable end-to-end training, it is necessary to compute the gradient of the EM iteration $\partial\lambda_{k+1}/\partial\lambda_k$. While this gradient exists, we will show in Section 3 that exactly computing this gradient is computationally expensive. A key technical contribution of this paper is in proving the gradient of the EM iteration can be reliably approximated via simultaneous perturbation stochastic approximation (SPSA) (Spall, 1992; Sadegh & Spall, 1998; Spall, 1998; Sadegh, 1997), which only involves function evaluation and can be easily computed. With such gradient approximation technique, we expect a large body of functions whose gradient is difficult to compute can now be incorporated as layers in deep neural networks, dramatically expanding the choice of network layers and increasing the flexibility of network architecture design. We name our method DENNL (**D**ifferentiable **E**M as **N**eural **N**etwork **L**ayer).

It is worth noting that a user's feature consists of a number of categorical and numerical fields. Instead of hand-crafting a dedicated neural network to encode the user feature, we used a pre-trained low-dimensional *user embedding* to represent such information. The user embedding was trained on a link prediction task using the *graph convolutional network* (GraphSage) (Hamilton et al., 2017; Cheng et al., 2016). Much similar to the success of pre-trained word embedding in downstream natural language processing tasks (Peters et al., 2018; Devlin et al., 2018), we demonstrate that the embedding trained using GraphSage is effective in summarizing information with complex structures and can be highly instructive in seemingly unrelated downstream tasks.

## 2 BACKGROUNDS

**Hidden Markov Model and EM Algorithm.** In this section, we only introduce the concepts and notations related to HMM and EM algorithm that are necessary for further discussion, and refer the readers to (Rabiner, 1989) and (Bilmes et al., 1998) for a more comprehensive treatment of the topics.

A HMM is defined by a set of hidden states, the transition probability of the hidden states, and the emission distribution. The Markov property mandates that the state transition and the emission distribution is only dependent on the current hidden state. The realizations of the emission distribution are what we can observe as an observation sequence. Let the distribution of the initial hidden state be $\pi$ and $Q_t, \{t = 1, \cdots, T\}$ be the hidden state at time step $t$. Then transition probability is defined by a time-homogeneous transition matrix $A$ with elements $a_{i,j} = P(Q_t = j|Q_{t-1} = i)$, where $i, j \in \{1, \cdots, N\}$ are the indices of the hidden states. Let $O_t$ be observable at time step $t$ and $o_t$ to be a realization of $O_t$. The emission distribution defines the probability of a particular observation conditioned on the current hidden state

$$b_j(o_t) = P(O_t = o_t|Q_t = j). \tag{1}$$

The emission probability of all hidden states can be represented by $B = \{b_j(.)\}$.

Now we follow the notation in (Bilmes et al., 1998) to present a brief introduction of EM algorithms applied in HMMs. A key concept is the forward and backward probability, which is used to compute the likelihood efficiently. Forward probability is the probability of seeing the partial sequence $o_1, \cdots, o_t$ and ending up in hidden state $i$ at time $t$, Define the forward probability as:

$$\alpha_i(t) = p(O_1 = o_1, \cdots, O_t = o_t, Q_t = i|\lambda),$$

where parameters $\lambda = (A, B, \pi) = \{\lambda_1, \cdots, \lambda_M\}$ and $M$ is the number of parameters. The forward probability can be computed efficiently in an iterative manner,

$$\alpha_i(1) = \pi_i b_i(o_1), \quad \alpha_j(t+1) = \left[\sum_{i=1}^{N} \alpha_i(t) a_{i,j}\right] b_j(o_{t+1}). \tag{2}$$

Similarly, Backward probability is the probability of the ending partial sequence $o_{t+1}, \cdots, o_T$ given the hidden state $i$ at time $t$ and is defined as:

$$\beta_i(t) = p(O_{t+1} = o_{t+1}, \cdots, O_T = o_T|Q_t = i, \lambda). \tag{3}$$

Similarly backward probability can also be computed iteratively:

$$\beta_i(T) = 1, \quad \beta_i(t) = \sum_{j=1}^{N} a_{i,j} b_j(o_{t+1}) \beta_j(t+1). \tag{4}$$

Given the observation sequence $o_1, \cdots, o_T$ and the parameter estimation at the $k$-th iteration $\lambda^{(k)}$, the parameter estimation at the $k + 1$-th iteration can be computed as follows:

$$\pi_i^{(k+1)} \quad = \quad \frac{\alpha_i(1)\beta_i(1)}{\sum_{t=1}^{N} \alpha_t(1)\beta_t(1)}, \tag{5}$$

$$a_{i,j}^{(k+1)} \quad = \quad \frac{\sum_{t=1}^{T-1} \alpha_i(t)a_{i,j}b_j(o_{t+1})\beta_j(t+1)}{\sum_{t=1}^{T-1} \alpha_i(t)\beta_i(t)}, \tag{6}$$

$$b_i^{(k+1)}(h) \quad = \quad \frac{\sum_{t=1}^{T} \alpha_i(t)\beta_i(t)\delta_{o_t,v_h}}{\sum_{t=1}^{T} \alpha_i(t)\beta_i(t)}, \tag{7}$$

where $\delta_{o_t,v_h}$ is an indicator function which equals to 1 if $o_t = v_h$ and 0 otherwise.

**Hidden Semi-Markov Model.** HSMM (Ostendorf et al., 1996) explicitly models the duration of a state, by introducing a *duration distribution*, with which multiple observations could be emitted from a hidden state before state transition occurs. A HSMM can be parameterized as HMM by introducing a counter of the time steps in the current state (Yildirim et al., 2013; Bietti et al., 2015). For this reason, the computation methods as well as the analysis of HMM naturally apply to HSMM.

## 3 PROBLEM STATEMENT AND PROPOSED SOLUTION

In this section we formally state the problem and describe our proposed method. The objective is to estimate the parameters $\{\lambda_{(u)}\}$ of a collection of HMM models, using the corresponding observation sequence $\{o_{t,1:T_{(u)}}\}$ and feature embedding $\{e_{(u)}\}$ as input. Here $(u)$ is the index of a HMM in the collection of HMMs, and $T_{(u)}$ is the length of the observation sequence. The observation sequence $\{o_{t,1:T_{(u)}}\}$ is noisy or insufficient for the conventional EM algorithm to apply. The feature embedding $e_{(u)}$ is a low-dimensional vector that encodes our prior knowledge of $\{\lambda_{(u)}\}$. We propose to use neural network to combine the information across all the HMMs for more effective parameter estimation.

$$\lambda_{(u)} = \Psi\left(e_{(u)}, o_{t,1:T_{(u)}}, \Phi\right) \tag{8}$$

where $\Phi$ is the parameter of the neural network that is shared by the HMM models. Formally, the loss function to be minimized is

$$\mathcal{L}(\Phi) = -\sum_{\forall u} L\left(\lambda_{(u)}, o_{t,1:T_{(u)}}\right) \tag{9}$$

where $L(\cdot)$ is the log-likelihood of observing sequence $o_{t,1:T_{(u)}}$ given HMM parameter $\lambda_{(u)}$.

The overview of our proposed method is shown in Figure 1. Our architecture can be considered as an unrolled Recurrent Neural Network (RNN), where each recurrence (or stage) of the RNN corresponds to an EM iteration. Our key novelty is in rendering EM iteration as a network layer. The resulted "EM layer" serves as the output function of the RNN. This is in contrast to the traditional RNN where the output function is a fully connected layer with non-linearity. Note that the EM layer is parameter-less and directly reflects the domain knowledge and design intent of network designer. As we will show in Section 6, such recurrent structure that mimics the iterative EM iterations in a conventional setting is critical to the success of our proposed method.

As in the conventional EM algorithm, an initial parameter is provided and is iteratively refined. After the $k$th EM iteration is performed, a multilayer perceptron (MLP) which is labeled as DNN in the figure transforms $\lambda^{(k)}$, the latent state from the previous iteration $h_k$, and the user embedding to an updated $\lambda^{(k+1)}$ and an updated latent state $h_{k+1}$. The parameters of the MLP are shared across EM iterations. It is worth emphasizing that the latent state $h$ captures the state of the DNN and shall not be confused with the hidden state of the HMMs. We use the `Softplus` function and $L_1$ normalization to ensure the elements of the resulted $\lambda$ are in proper range.

**Differentiability of the EM iteration**. The EM iterations shown in Figure 1 can be considered as layers of a neural network, with $\lambda^{(k)}$ and the observation sequence as input and $\lambda^{(k+1)}$ as output. To enable end-to-end training, it is necessary to compute the partial gradient $\partial\lambda^{(k+1)}/\partial\lambda^{(k)}$. Even though the EM iteration for HMM can be explicitly stated (Equation 5), directly computing the partial gradient $\partial\lambda^{(k+1)}/\partial\lambda^{(k)}$ using a automatic differentiation engine faces two numerical challenges, both of which are related to the forward and backward probability (Equation 2 and 4).

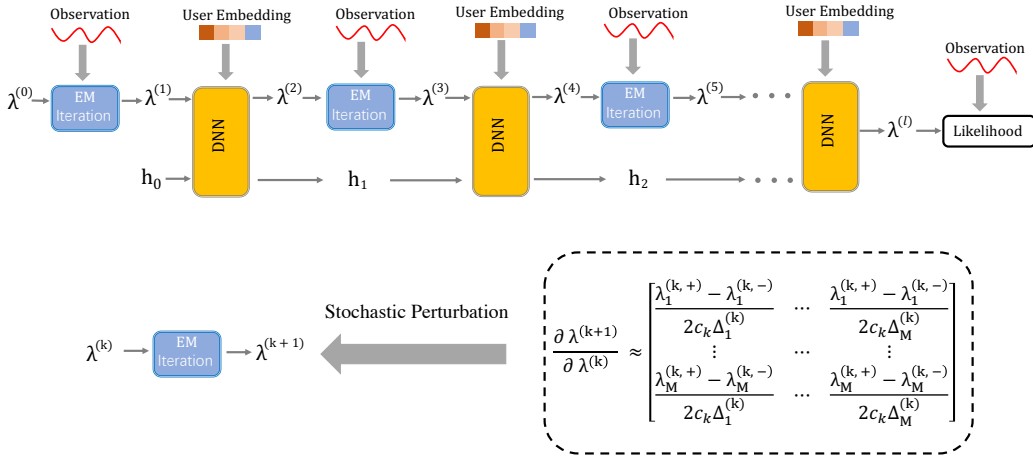

Figure 1: Overview of DENNL. The EM iteration are incorporated in a deep neural network as layers. End-to-end training is enabled by approximating the gradient of the "EM layer" using SPSA, as shown in the low half of the figure.

- While evaluating the alpha-beta equation is a $\mathcal{O}(T)$ operation where $T$ is the length of the observation sequence, the number of nodes required to build the forward and backward probability in the *computation graph* of an automatic differentiation engine is on the order of $\mathcal{O}(T^2)$. Empirically we found this leads to intractable computation cost.
- The length of the observation sequence $T$ varies across different HMM models, thus a separate computation graph for the forward and backward probability has to be rebuilt for every distinct $T$ value. This is particularly problematic when the HMMs in a mini-batch comes with different $T$ values.

To address the aforementioned numerical challenges, we propose to approximate the gradient $\partial \lambda^{(k+1)}/\partial \lambda^{(k)}$ using SPSA. Much similar to how we can approximate the gradient of a one-dimensional function using the finite difference method, SPSA estimates the gradient of a function by evaluating the function with stochastically perturbed input vectors. A series of work by Spall and colleagues (Spall, 1992; Sadegh & Spall, 1998; Spall, 1998; Sadegh, 1997) established theoretically rigorous error bound for certain scalar functions. Note that such gradient approximation only involves evaluating the EM iteration and can be efficiently computed.

A important question to answer is how well can SPSA approximate the gradient $\partial \lambda^{(k+1)}/\partial \lambda^{(k)}$ in our specific application. In the next section, we formally describe how SPSA is performed and establish its convergence property for EM iterations.

## 4  SPSA AND CONVERGENCE ANALYSIS

As the name suggests, Simultaneous Perturbation Stochastic Approximation approximates the gradient (Jacobian) of a function by simultaneously applying a small perturbation to all the dimensions of the argument of the function. Let $\Delta \in \mathcal{R}^M$ be a $M$-dimensional random vector, where elements $\Delta_1, \cdots, \Delta_M$ are independent with zero mean. We do not assume specific distribution of $\Delta$ as long as $E(1/\Delta)$ is bounded. We treat the EM iteration stated in equation 5, equation 6 and equation 7 as a function, with the HMM parameter at the $k$-th iteration $\lambda^{(k)}$ being the function argument and the refined parameter at $(k+1)$-th iteration $\lambda^{(k+1)}$ as the output:

$$\lambda^{(k+1)} = \mathbf{f}(o, \lambda^{(k)}), \tag{10}$$

where $\mathbf{f} : \mathcal{R}^{\mathbf{M}} \to \mathcal{R}^{\mathbf{M}}$ and $o = \{o_1, \cdots, o_T\}$ is the observation sequence.

The SPSA estimates the gradient (Jacobian) of a function through two function evaluations. The arguments of the two function evaluation are positively and negatively perturbed respectively as

following:

$$\lambda^{(k,+)} \quad = \quad \mathbf{f}\left(\lambda^{(k)} + c_k\Delta^{(k)}\right) + \epsilon^{(k,+)} \tag{11}$$

$$\lambda^{(k,-)} \quad = \quad \mathbf{f}\left(\lambda^{(k)} - c_k\Delta^{(k)}\right) + \epsilon^{(k,-)} \tag{12}$$

where $c_k$ is a positive scalar and $\epsilon^{(k,+)}, \epsilon^{(k,-)}$ are measurement noise. Let $\mathcal{F}_k$ be the random field generated by $\lambda^{(0)}, \cdots, \lambda^{(k)}$. We assume the measure noise satisfy the following conditions:

$$E\left(\epsilon^{(k,+)} - \epsilon^{(k,-)} \middle| \mathcal{F}_k, \Delta_k\right) = \mathbf{0}, \tag{13}$$

where $\mathbf{0}$ is the $M$-dimensional zero column vector.

With $\lambda^{(k,+)}$ and $\lambda^{(k,-)}$, the approximation of the true gradient (Jacobian) $\mathbf{J}^{(k)}$ due to SPSA is defined as

$$\hat{\mathbf{J}}^{(k)} = \begin{bmatrix} \frac{\lambda_1^{(k,+)} - \lambda_1^{(k,-)}}{2c_k\Delta_1^{(k)}} & \cdots & \frac{\lambda_1^{(k,+)} - \lambda_1^{(k,-)}}{2c_k\Delta_M^{(k)}} \\ \vdots & \cdots & \vdots \\ \frac{\lambda_M^{(k,+)} - \lambda_M^{(k,-)}}{2c_k\Delta_1^{(k)}} & \cdots & \frac{\lambda_M^{(k,+)} - \lambda_M^{(k,-)}}{2c_k\Delta_M^{(k)}} \end{bmatrix}, \tag{14}$$

where $\lambda_i^{(k,+)}, \lambda_i^{(k,-)}, \Delta_i^{(k)}$ are the $i$-th element of $\lambda^{(k,+)}, \lambda^{(k,-)}, \Delta^{(k)}$, respectively.

We now quantify the approximation error of $\hat{\mathbf{J}}^{(k)}$ and establish the convergence property of the gradient approximation due to SPSA for the EM iteration. Our primary theoretical contribution are Lemma 1 and Theorem 1, which are stated below. Note that (Spall, 1992) offered similar theoretical results but their results only apply to functions with scalar output. Our proof took a very different approach from the proof in (Spall, 1992) to bound the Frobenius Norm of the approximated Jacobian to account for the fact that EM iteration is a function with vector input and vector output ($\mathbf{f} : \mathcal{R}^M \to \mathcal{R}^M$). Please see Appendix A for the detailed proof.

**Lemma 1.** *For $i$-th function $f_i$ of $\mathbf{f}$ in equation 10, $\frac{\partial^3 f_i}{\partial\lambda_{j_1}\partial\lambda_{j_2}\partial\lambda_{j_3}}$ exist and are uniformly bounded on the ranges of parameter $\lambda$ such that for $i, j_1, j_2, j_3 = 1, \cdots, M$,*

$$\left|\frac{\partial^3 f_i}{\partial\lambda_{j_1}\partial\lambda_{j_2}\partial\lambda_{j_3}}(o, \lambda)\right| \leq \tau_0, \tag{15}$$

*where $o$ is the observation sequence and $\tau_0$ is a positive scalar.*

**Theorem 1.** *Suppose that for each $k$, $\Delta_1^{(k)}, \cdots, \Delta_M^{(k)}$ are independent to each other and have zero mean such that $E(\Delta_i^{(k)}) = 0$, $i = 1, 2, \cdots, M$. In addition, as $k \to \infty$, almost surely $|\Delta_i^{(k)}| \leq \tau_1$ and $E|\frac{1}{\Delta_i^{(k)}}| \leq \tau_2$, $i = 1, \cdots, M$, where $\tau_1$ and $\tau_2$ are positive constants. Then for $k \to \infty$, almost surely*

$$\left\|E\left(\hat{\mathbf{J}}^{(k)} - \mathbf{J}^{(k)} \middle| \mathcal{F}_k\right)\right\|_F = O(c_k^2), \tag{16}$$

*where $\|.\|_F$ is Frobenius norm.*

*Sketch of Proof:* The difference between the $i, j$-th element of the approximated Jacobian $\hat{\mathbf{J}}^{(k)}$ and the true Jacobian $\mathbf{J}^{(k)}$ is denoted by $\hat{\mathbf{J}}_{i,j}^{(k)} - \mathbf{J}_{i,j}^{(k)}$. We compute the third order Taylor series expansion (Taylor & Lay, 1958) of this difference, and establish that the first two order terms vanish. The constant term is obviously zero. The first order term disappears because the elements of $\Delta_k$ are independent with zero mean. The second order term is also zero because the positive perturbation and the negative perturbation cancel out. According to Lemma 1, the third order term is uniformly bounded, which guarantees the convergence rate of SPSA for estimating the gradient of the EM iteration.

## 5 RELATED WORK

It is shown in (Wen et al., 2017) that even a crude approximation of the gradient is sufficient for stochastic gradient descent to converge. Such theoretical result is among our motivations to explore

gradient approximation techniques for layers whose gradient is expensive to compute. There are a number of works on enhancing classical sequential data analysis methods with deep neural networks. Krishnan (Krishnan et al., 2017) employs a RNN to parameterize a variational approximation of the posterior distribution in a Gaussian state space model. Other works along this line include (Bui et al., 2016), (Salinas et al., 2017) and (Linderman et al., 2017).

Perhaps the most related existing work is (Rangapuram et al., 2018), where the authors used a deep neural network to parameterize a collection of linear state space models so the information across multiple time series can be shared. The underlying linear state space model is updated via Kalman Filtering, which only involves matrix-to-matrix multiplication and is differentiable. The price to pay for such differentiability is the underlying sequential data analysis method has to be a relative simple form (e.g. linear state space model) with limited expressiveness and flexibility. In contrast, in this work we do not have to limit ourselves to methods whose gradient are easy to compute.

Another closely related work is Gref *et al.* (Greff et al., 2017), where the author incorporates the so-called *general EM algorithm* (Wu, 1983) in the neural network for unsupervised clustering. The key difference is that instead of performing a complete EM update, the general EM algorithm (Greff et al., 2017) only takes one step of gradient descent in the M step of the EM algorithm to refine the parameter estimation. It may take a much larger number of iterations for the general EM algorithm to converge (Wu, 1983). Larger number of iterations leads to a deeper neural network, as an additional layer is stacked in the neural network per iteration. A network that is too deep may cause difficulty in training (He et al., 2016). Hinton *et al.* (Hinton et al., 2018) directly computes the gradient of an EM based routing procedure. With a small number of EM iterations, (Hinton et al., 2018) does not face the numerical challenge outlined in Section 3 because the sizes of their inputs in a mini-batch are fixed and the problem is of much smaller scale.

**Relationship with other recommender systems algorithms.** Graph embedding (Hamilton et al., 2017; Grbovic & Cheng, 2018) is one of the backbone algorithms for today's industrial recommender systems. While graph embedding is highly effective in capturing complex interactions among hundreds of millions of users and items, most of the algorithms fail to consider the temporal evolvement of the user's interest (with a few exceptions such as (Kumar et al., 2019), (Trivedi et al., 2017) and (Zhou et al., 2018), none of which provides an *interpretable model* of user interest evolvement). We do not consider our proposed method as a general solution for item level recommendation, as it is unlikely that our proposed method in its current form can be scaled to millions of items. Rather, the focus of our proposed method is to capture highly detailed temporal interest evolvement of the users across a limited number of categories.

## 6 EXPERIMENT

In this section, we evaluate the effectiveness of our proposed method on synthetic data as well as real-world e-Commerce data.

The primary contribution of our proposed method is to share the information across multiple HMMs for parameter estimation in cases where the observation sequence is noisy or insufficient. To the best of our knowledge, there is no directly comparable method in the existing literature. We perform ablation studies against the following baselines to justify our design choices:

1. **Conventional EM**. The HSMM parameters are estimated by the conventional EM algorithm, with no information sharing across the HSMM models.

2. **EM Mapping**. We first estimate the HSMM parameters using the conventional EM algorithm, then train a neural network (a stack of fully connected layers) that maps the user embedding to the HSMM parameters due to conventional EM algorithm. The training objective is to minimize the square error. Though the neural network establishes some correlation across the HSMM models, the HSMM parameters due to the conventional EM algorithm solely relies on the user's corresponding observation sequence that is noisy and/or insufficient.

3. **MLP Mapping**. In this baseline, a multilayer perceptron (MLP) maps the user embedding to the HMM parameters, then the likelihood is computed based on the HMM parameters and the observation sequence. We use gradient ascent to maximize the likelihood. Note that this baseline is purely based on gradient ascent and does not involve the EM algorithm.

4. **Initial Fusion**. A DNN maps user embedding to HMM parameters, which are refined by the EM iterations. Here we only fuse the observations across different HMMs at the initial stage. In a sense, this architecture identifies an optimal initial parameter guess. In contrast, our proposed method fuses observations across different HMMs at every EM iteration.

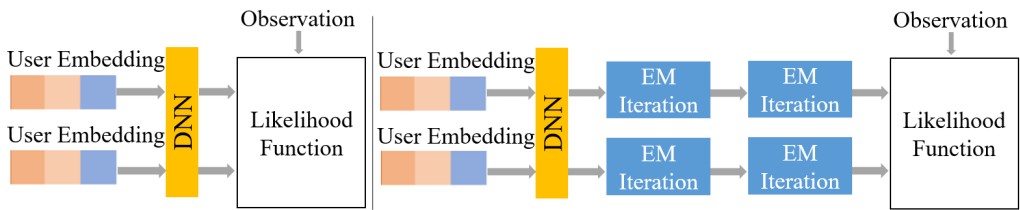

Figure 2: Overview of baselines. Left: MLP Mapping. Right: Initial Fusion.

## 6.1 SYNTHETIC DATA

The purpose of evaluating the proposed method on synthetic data is to compare the decoded hidden state with known ground truth state. Being able to correctly infer the hidden state is a key metric of HMM models (Sarkar & Dunson, 2018). We pre-designed the parameters of a collection HSMM, each of which has 10 hidden states. The emission distribution is a 5-dimension Gaussian distribution. The duration distribution is Poisson distribution. To generate an informative feature embedding for each HSMM, we serialize the HSMM parameters as a vector and apply a non-linear transformation (in the form of $Ax + b$ followed by ReLU, where $x$ is the parameter vector, $A$ is a known full rank matrix and $b$ is a known vector). The resulted feature embedding is strongly related to its underlying parameter, but one cannot recover the HMM parameters from its feature embedding. The evaluation metric is the percentage of the decoded states (Viterbi, 1967; Munkres, 1957) that match the ground truth state, or the labeling accuracy.

We first evaluate our method in cases where the observation sequence is insufficient. We also compared DENNL to baseline methods when a certain percentage of the observations in the observation sequence generated by the ground truth model is contaminated by noise. If a particular observation is chosen to be noisy, its value is replaced by the value of another randomly chosen observation in the sequence. The results are reported in Table 1.

Table 1: Labeling accuracy comparison of DENNL to the baseline methods on the synthetic data.

| Method | Sequence Length: 50 | Sequence Length: 100 | Sequence Length: 400 | Noisy Sample: 20% | Noisy Sample: 10% |
|---|---|---|---|---|---|
| Conventional EM | 82.0% | 85.5% | **94.1%** | 83.1% | 91.5% |
| Direct Mapping | 71.4% | 72.2% | 73.0% | 67.9% | 71.4% |
| MLP Mapper | 51.5% | 50.3% | 52.1% | 46.2% | 48.5% |
| Initial Fusion | 82.4% | 83.7% | 92.6% | 82.8% | 89.7% |
| DENNL | **88.3%** | **90.0%** | 93.5% | **88.1%** | **92.3%** |

**Discussion:** As shown in Table 1, DENNL consistently outperforms the baseline methods with noisy or insufficient observations. When the observation sequence is of sufficient length or noise-less, the benefit of the proposed method diminishes as expected. Even though Direct Mapping method enables some information sharing across HMMs, its performance was poor because the regression target is due to the conventional EM and can be inaccurate to begin with. This experiment illustrates the necessity to fuse information across HMM in every EM iteration. While the Initial Fusion method seems to enable information sharing, its performance is very close to that of conventional EM. This method essentially attempts to identify an optimal initial solution with shared information across HMMs. However, at least in our application, the EM algorithm appears to be insensitive to the initial solution. The MLP Mapper method enables information sharing with simple architecture, but is

slow to converge and is not competitive even against conventional EM. To explain this, note that in a conventional setting, we seldom use gradient ascent for HMM parameter estimation. Rather, the common practice is to use the EM algorithm which is carefully designed for the particular application.

***This is the key message that we would like to convey to the community***: operations that are inspired by well-studied classical methods (such as EM iteration) can be significantly more effective than operations that are synthesized using generic layers (convolution, fully connected, etc). If the gradient of the classical method is incomputable, it could potentially be numerically approximated. This formulation dramatically expands the choices of network layers and allows network designers to directly apply their domain knowledge in the network architecture.

## 6.2   E-COMMERCE DATA

As previously discussed in Section 5, DENNL is not considered as a general solution for recommendation systems in its current form. Rather, it is suitable to model the shift of user's interest across a small number of categories. We apply DENNL in a clearly defined and fast-growing sector on our e-Commerce platform, namely Home Decoration. The Home Decoration sector comes with certain well-defined shopping patterns that follow the progress of the decoration of a newly purchased condo. Formally, we model the behavior of the user in the Home Decoration category by HSMM with 17 states, each of which corresponds to a sub-category in the Home Decoration sector. Please see Appendix A.2 for more information of the e-Commerce dataset.

In this setting we no longer have access to the ground truth states, thus we can no longer use labeling accuracy as the metric. Instead, we use a link prediction metric. Using the trained HSMM model, we could compute the likelihood of a user interacting with each of the 17 sub-categories. The prediction is considered to be successful if the user interacted with the predicted sub-category with the highest likelihood, top-3 and top-5 likelihood.

Table 2: Sub-cateogry prediction accuracy in the e-Commerce dataset.

| Method | Top-1 | Top-3 | Top-5 |
|---|---|---|---|
| Conventional EM | 10.3% | 23.9% | 50.5% |
| GraphSage (Hamilton et al., 2017) | 17.7% | 37.4% | 65.5% |
| MLP Mapper | 7.4% | 19.6% | 35.1% |
| Initial Fusion | 11.5% | 21.6% | 48.2% |
| DENNL | **21.8%** | **48.5%** | **75.2%** |

Our proposed method outperforms the conventional EM as well as the GraphSage (Hamilton et al., 2017) in this experiment. Note that the user feature we used is the user embedding due to GraphSage, thus DENNL has all the information that GraphSage can offer. One way to interpret the result is that our method is enhancing GraphSage with an HSMM that captures the temporal interest shift of the users.

## 7   CONCLUSION

In this work, we proposed a novel method in combining EM algorithm with neural network to predict the parameters of a collection of Hidden Markov Models. The basic idea is to use the neural network to capture correlation among the HMMs so the information across different HMMs can be shared and re-used. The key technical contribution of this work is in providing a proof that the gradient of the EM iteration can be efficiently and reliably approximated using the SPSA. We evaluated our proposed on synthetic data as well as real-world e-Commerce data. An interesting observation is that the user embedding trained in an unrelated task using graph convolutional neural network can be highly informative in HMM parameter estimation. Parallels can be drawn between this observation and the success of pre-trained word embedding in downstream natural language processing tasks.

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

## A APPENDIX

### A.1 PROOF OF LEMMA 1 AND THEOREM 1

**Lemma 1.** For $i$-th dimension $f_i$ of $\mathbf{f}$ in equation 10, $\frac{\partial^3 f_i}{\partial \lambda_{j_1} \partial \lambda_{j_2} \partial \lambda_{j_3}}$ exist and are uniformly bounded on the range of parameter $\lambda$ such that for $i, j_1, j_2, j_3 = 1, \cdots, M$,

$$\left| \frac{\partial^3 f_i}{\partial \lambda_{j_1} \partial \lambda_{j_2} \partial \lambda_{j_3}}(o, \lambda) \right| \leq \tau_0, \tag{17}$$

where $o$ is the observation sequence and $\tau_0$ is a positive scalar.

*Proof.* We first show that that the numerator and denominator of $f_i$ is the polynomial function of $M$-parameters. Since the range of parameters are closed and bounded set, then it follows (Rudin et al., 1964) that the third partial gradient of $f_i$ is uniformly continuous and up bounded and then Lemma 1 is established.

Rewrite equation 2 as

$$[\alpha_1(t+1), \cdots, \alpha_N(t+1)] = [\alpha_1(t), \cdots, \alpha_N(t)] \times \Sigma^{(t+1)} \tag{18}$$

where

$$\Sigma^{(t+1)} = A \times \begin{bmatrix} b_1(o_{t+1}) & & \\ & \ddots & \\ & & b_N(o_{t+1}) \end{bmatrix}. \tag{19}$$

Then it follows that

$$[\alpha_1(t), \cdots, \alpha_N(t)] = [\alpha_1(1), \cdots, \alpha_N(1)] \times \prod_{l=2}^{t} \Sigma^{(l)}, \tag{20}$$

where $t = 1, \cdots, T$. For $t = 1$, let $\prod_{l=2}^{t} \Sigma^{(l)} = I_N$ where $I_N$ is a $N \times N$ identity matrix. According to equation 18, we have

$$\alpha_i(t) = [\alpha_1(1), \cdots, \alpha_N(1)] \times \prod_{l=2}^{t} \Sigma^{(l)} \times e_i, \tag{21}$$

where $e_i$ is $N$-column vector, the $i$-th element equals to one and the rest elements equal to zeros.

Rewrite equation 4 as

$$\begin{bmatrix} \beta_1(t) \\ \vdots \\ \beta_N(t) \end{bmatrix} = \Sigma^{(t+1)} \times \begin{bmatrix} \beta_1(t+1) \\ \vdots \\ \beta_N(t+1) \end{bmatrix},$$

which together with $\beta_i(T) = 1$, $i = 1, \cdots, N$, yields that

$$\begin{bmatrix} \beta_1(t) \\ \vdots \\ \beta_N(t) \end{bmatrix} = \prod_{l=t+1}^{T} \Sigma^{(l)} \times \mathbf{1}_N, \tag{22}$$

where $\mathbf{1}_N$ is $N$-dimensional column vector with all elements equaling to one, $t = 1, \cdots, T$. For $t = T$, let $\prod_{l=t+1}^{T} \Sigma^{(l)} = I_N$ . According to equation 22, we have

$$\beta_i(t) = e_i^T \times \prod_{l=t+1}^{T} \Sigma^{(l)} \times \mathbf{1}_N,$$

which together with equation 21 yields that

$$\alpha_i(t)\beta_i(t) = [\alpha_1(1), \cdots, \alpha_N(1)] \times \prod_{l=2}^{t} \Sigma^{(l)} \times e_i e_i^T \times \prod_{l=t+1}^{T} \Sigma^{(l)} \times \mathbf{1}_N. \tag{23}$$

Define $N$-dimensional column vector as

$$V^{(i,t)} = \prod_{l=2}^{t} \Sigma^{(l)} \times e_i e_i^T \times \prod_{l=t+1}^{T} \Sigma^{(l)} \times \mathbf{1}_N. \tag{24}$$

Let $m_{i_l,j_l}$ be the $i_l, j_l$-th element of $\Sigma^{(l)}$. Then the $\ell$-th element of $V_\ell^{(i,t)}$ is just linear combination of elements $m_{i_2,j_2} \cdots m_{i_T,j_T}$, where the coefficient in front of $m_{i_2,j_2} \cdots m_{i_T,j_T}$ is defined as $c_{i_2,j_2,\cdots,i_T,j_T}^{(i,t,\ell)}$. It follows that

$$V^{(i,t)} = \begin{bmatrix} \sum_{i_2,j_2,\cdots,i_T,j_T} c_{i_2,j_2,\cdots,i_T,j_T}^{(i,t,1)} m_{i_2,j_2} \cdots m_{i_T,j_T} \\ \vdots \\ \sum_{i_2,j_2,\cdots,i_T,j_T} c_{i_2,j_2,\cdots,i_T,j_T}^{(i,t,N)} m_{i_2,j_2} \cdots m_{i_T,j_T} \end{bmatrix}. \tag{25}$$

Combining equation 23, equation 24 and equation 25, we have

$$\alpha_i(t)\beta_i(t) = \sum_{\ell=1}^{N} \sum_{i_2,j_2,\cdots,i_T,j_T} \pi_\ell b_\ell(o_1) c_{i_2,j_2,\cdots,i_T,j_T}^{(i,t,\ell)} m_{i_2,j_2} \cdots m_{i_T,j_T}. \tag{26}$$

According to equation 19, given that observation $o_l = h$, $m_{i_l,j_l} = a_{i_l,j_l} b_{j_l}(o_l) = a_{i_l,j_l} b_{j_l}(h)$. From equation 26, it is clearly that $\alpha_i(t)\beta_i(t)$ is the polynomial function of parameters $\{A, B, \pi\}$. From equation 5 and equation 26, we have

$$\pi_i^{(k+1)} = \frac{\sum_{\ell=1}^{N} \sum_{i_2,j_2,\cdots,i_T,j_T} \pi_\ell b_\ell(o_1) c_{i_2,j_2,\cdots,i_T,j_T}^{(i,1,\ell)} m_{i_2,j_2} \cdots m_{i_T,j_T}}{\sum_{i=1}^{N} \sum_{\ell=1}^{N} \sum_{i_2,j_2,\cdots,i_T,j_T} \pi_\ell b_\ell(o_1) c_{i_2,j_2,\cdots,i_T,j_T}^{(i,1,\ell)} m_{i_2,j_2} \cdots m_{i_T,j_T}}. \tag{27}$$

The numerator and denominator of $\pi_i^{(k+1)}$ are polynomial function of parameters $\lambda = \{A, B, \pi\}$. According to (Rudin et al., 1964), the third partial gradient of $\pi_i^{(k+1)}$ to $\lambda$ is uniformly bounded on the range of parameters, which is bounded and closed set on $\mathcal{R}^M$.

Since $a_{i,j} b_j(o_{t+1}) = m_{i_t,j_t}$, where $i_t = i$, $j_t = j$, then similar as equation 26, we have

$$\alpha_i(t) a_{i,j} b_j(o_{t+1}) \beta_j(t+1) = \alpha_i(t) m_{i_t,j_t} \beta_j(t+1)$$

$$= \sum_{\ell=1}^{N} \sum_{i_2,j_2,\cdots,i_T,j_T} \pi_\ell b_\ell(o_1) \tilde{c}_{i_2,j_2,\cdots,i_T,j_T}^{(i,j,t,\ell)} m_{i_2,j_2} \cdots m_{i_T,j_T}, \tag{28}$$

where index $i, j, t$ of coefficients $\tilde{c}_{i_2,j_2,\cdots,i_T,j_T}^{(i,j,t,\ell)}$ corresponds to $i, j, t$ in $\alpha_i(t) a_{i,j} b_j(o_{t+1}) \beta_j(t+1)$. Combining equation 6, equation 26 and equation 28, we have

$$a_{i,j}^{(k+1)} = \frac{\sum_{t=1}^{T-1} \sum_{\ell=1}^{N} \sum_{i_2,j_2,\cdots,i_T,j_T} \pi_\ell b_\ell(o_1) \tilde{c}_{i_2,j_2,\cdots,i_T,j_T}^{(i,j,t,\ell)} m_{i_2,j_2} \cdots m_{i_T,j_T}}{\sum_{t=1}^{T-1} \sum_{\ell=1}^{N} \sum_{i_2,j_2,\cdots,i_T,j_T} \pi_\ell b_\ell(o_1) c_{i_2,j_2,\cdots,i_T,j_T}^{(i,t,\ell)} m_{i_2,j_2} \cdots m_{i_T,j_T}}. \tag{29}$$

Similarly, it follows from equation 7 and equation 26 that

$$b_i^{(k+1)}(h) = \frac{\sum_{t=1}^{T} \sum_{\ell=1}^{N} \sum_{i_2,j_2,\cdots,i_T,j_T} \pi_\ell b_\ell(o_1) \delta_{o_t,v_h} c_{i_2,j_2,\cdots,i_T,j_T}^{(i,t,\ell)} m_{i_2,j_2} \cdots m_{i_T,j_T}}{\sum_{t=1}^{T} \sum_{\ell=1}^{N} \sum_{i_2,j_2,\cdots,i_T,j_T} \pi_\ell b_\ell(o_1) c_{i_2,j_2,\cdots,i_T,j_T}^{(i,t,\ell)} m_{i_2,j_2} \cdots m_{i_T,j_T}}. \quad (30)$$

Note that the numerator and denominator of $a_{i,j}^{(k+1)}$ and $b_i^{(k+1)}(h)$ are also polynomial function of parameters $\lambda = \{A, B, \pi\}$. Then the third partial gradient $a_{i,j}^{(k+1)}$ and $b_i^{(k+1)}(h)$ are also bounded on the range of parameters. Since $\mathbf{f}$ in equation 10 is equivalent to equation 27, equation 29 and equation 30, the first partial gradient of $\mathbf{f}$ is uniformly bounded.

$\square$

**Theorem 1.** Suppose that for each $k$, $\Delta_1^{(k)}, \cdots, \Delta_M^{(k)}$ are independent to each other and have zero mean such that $E(\Delta_i^{(k)}) = 0$, $i = 1, 2, \cdots, M$. In addition, as $k \to \infty$, almost surely $|\Delta_i^{(k)}| \le \tau_1$ and $E|\frac{1}{\Delta_i^{(k)}}| \le \tau_2$, $i = 1, \cdots, M$, where $\tau_1$ and $\tau_2$ are positive constants. Then for $k \to \infty$, almost surely

$$\left\| E\left( \hat{\mathbf{J}}^{(k)} - \mathbf{J}^{(k)} \,\middle|\, \mathcal{F}_k \right) \right\|_F = O(c_k^2), \quad (31)$$

where $\|.\|_F$ is Frobenius norm.

*Proof.* According to equation 11, equation 12, and equation 14, the $i$-th row, $j$-th column element difference between $\hat{\mathbf{J}}^{(k)}$ and $\mathbf{J}^{(k)}$ is

$$\hat{\mathbf{J}}_{i,j}^{(k)} - \mathbf{J}_{i,j}^{(k)} = \frac{f_i\left(\lambda^{(k)} + c_k\Delta^{(k)}\right) - f_i\left(\lambda^{(k)} - c_k\Delta^{(k)}\right)}{2c_k\Delta_j^{(k)}} - \frac{\partial f_i(\lambda^{(k)})}{\partial \lambda_j} + \frac{\epsilon_i^{(k,+)} - \epsilon_i^{(k,-)}}{2c_k\Delta_j^{(k)}}. \quad (32)$$

Taylor series expansion (Taylor & Lay, 1958) of $f_i(.)$ is represented as

$$\begin{aligned} f_i\left(\lambda_k + c_k\Delta^{(k)}\right) &= f_i(\lambda^{(k)}) + \sum_{j_1=1}^{M} c_k\Delta_{j_1}^{(k)} \frac{\partial f_i(\lambda^{(k)})}{\partial \lambda_{j_1}} + \frac{1}{2}\sum_{j_1=1}^{M}\sum_{j_2=1}^{M} c_k^2\Delta_{j_1}^{(k)}\Delta_{j_2}^{(k)} \frac{\partial^2 f_i(\lambda^{(k)})}{\partial \lambda_{j_1}\partial \lambda_{j_2}} \\ &+ \frac{1}{6}\sum_{j_1=1}^{M}\sum_{j_2=1}^{M}\sum_{j_3=1}^{M} c_k^3\Delta_{j_1}^{(k)}\Delta_{j_2}^{(k)}\Delta_{j_3}^{(k)} \frac{\partial^3 f_i\left(\lambda^{(k)} + \xi^{(+)}\Delta^{(k)}\right)}{\partial \lambda_1 \partial \lambda_2 \partial \lambda_3}, \end{aligned}$$

and

$$\begin{aligned} f_i\left(\lambda_k - c_k\Delta^{(k)}\right) &= f_i(\lambda^{(k)}) - \sum_{j_1=1}^{M} c_k\Delta_{j_1}^{(k)} \frac{\partial f_i(\lambda^{(k)})}{\partial \lambda_{j_1}} + \frac{1}{2}\sum_{j_1=1}^{M}\sum_{j_2=1}^{M} c_k^2\Delta_{j_1}^{(k)}\Delta_{j_2}^{(k)} \frac{\partial^2 f_i(\lambda^{(k)})}{\partial \lambda_{j_1}\partial \lambda_{j_2}} \\ &- \frac{1}{6}\sum_{j_1=1}^{M}\sum_{j_2=1}^{M}\sum_{j_3=1}^{M} c_k^3\Delta_{j_1}^{(k)}\Delta_{j_2}^{(k)}\Delta_{j_3}^{(k)} \frac{\partial^3 f_i\left(\lambda^{(k)} - \xi^{(-)}\Delta^{(k)}\right)}{\partial \lambda_1 \partial \lambda_2 \partial \lambda_3}, \end{aligned}$$

where $\xi^{(+)}, \xi^{(-)} \in (0, c_k)$. Then it follows that

$$\frac{f_i\left(\lambda^{(k)} + c_k\Delta^{(k)}\right) - f_i\left(\lambda^{(k)} - c_k\Delta^{(k)}\right)}{2c_k\Delta_j^{(k)}} - \frac{\partial f_i(\lambda^{(k)})}{\partial \lambda_j} = \frac{1}{\Delta_j^{(k)}} \sum_{j_1 \ne j} \Delta_{j_1}^{(k)} \frac{\partial f_i(\lambda^{(k)})}{\partial \lambda_{j_1}} + RES, \quad (33)$$

where

$$RES = \frac{c_k^2}{12\Delta_j^{(k)}} \sum_{j_1=1}^{M}\sum_{j_2=1}^{M}\sum_{j_3=1}^{M} \Delta_{j_1}^{(k)}\Delta_{j_2}^{(k)}\Delta_{j_3}^{(k)} \left[ \frac{\partial^3 f_i\left(\lambda^{(k)} + \xi^{(+)}\Delta^{(k)}\right)}{\partial \lambda_1 \partial \lambda_2 \partial \lambda_3} + \frac{\partial^3 f_i\left(\lambda^{(k)} - \xi^{(-)}\Delta^{(k)}\right)}{\partial \lambda_1 \partial \lambda_2 \partial \lambda_3} \right].$$

Since $E\left(\Delta_{j_1}^{(k)}\right) = 0$, then

$$E\left( \frac{1}{\Delta_j^{(k)}} \sum_{j_1 \ne j} \Delta_{j_1}^{(k)} \frac{\partial f_i(\lambda^{(k)})}{\partial \lambda_{j_1}} \,\middle|\, \mathcal{F}_k \right) = E\left( \frac{1}{\Delta_j^{(k)}} \right) \sum_{j_1 \ne j} \frac{\partial f_i(\lambda^{(k)})}{\partial \lambda_{j_1}} E\left( \Delta_{j_1}^{(k)} \right) = 0. \quad (34)$$

Since $|\frac{\partial^3 f_i(.)}{\partial \lambda_1 \partial \lambda_2 \partial \lambda_3}| \leq \tau_0$ from Lemma 1, almost surely $|\Delta_j^{(k)}| \leq \tau_1$ and $E|\frac{1}{\Delta_j^{(k)}}| < \tau_2$, as $k \to \infty$, then given $\mathcal{F}_k$, the expectation of $RES$ in equation 33 is bounded such that as $k \to \infty$, almost surely

$$|E(RES|\mathcal{F}_k)| \leq \frac{\tau_0 c_k^2}{6} \sum_{j_1=1}^{M} \sum_{j_2=1}^{M} \sum_{j_3=1}^{M} E\left|\frac{\Delta_{j_1}^{(k)}\Delta_{j_2}^{(k)}\Delta_{j_3}^{(k)}}{\Delta_j^{(k)}}\right| \leq \frac{\tau_0 \tau_1^3 \tau_2 M^3}{6} c_k^2. \tag{35}$$

Combining equation 33, equation 34 and equation 35, almost surely

$$E\left(\frac{f_i\left(\lambda^{(k)} + c_k\Delta^{(k)}\right) - f_i\left(\lambda^{(k)} - c_k\Delta^{(k)}\right)}{2c_k\Delta_j^{(k)}} - \frac{\partial f_i(\lambda^{(k)})}{\partial \lambda_j}\middle| \mathcal{F}_k\right) = O(c_k^2), \tag{36}$$

In addition, from equation 13, we have

$$E\left(\frac{\epsilon_i^{(k,+)} - \epsilon_i^{(k,-)}}{2c_k\Delta_j^{(k)}}\middle| \mathcal{F}_k\right) = 0$$

which together with equation 32 and equation 36 yields that as $k \to \infty$, almost surely

$$E\left(\hat{\mathbf{J}}_{i,j}^{(k)} - \mathbf{J}_{i,j}^{(k)}\middle| \mathcal{F}_k\right) = O(c_k^2),$$

It follows that

$$\left\|E\left(\hat{\mathbf{J}}^{(k)} - \mathbf{J}^{(k)}\middle| \mathcal{F}_k\right)\right\|_F = O(c_k^2).$$

$\square$

## A.2    DESCRIPTION OF E-COMMERCE DATASET

The operation specialist who has deep domain knowledge in the Home Decoration sector have concluded that after a condo is purchased, the customer will likely go through 4 distinct phases: `Planning`, `Indoor Construction`, `Furniture`, and `Accessories`. In the `Planning` phase the customer mainly browses listing of designing services and contractors. Representative items in the `Indoor Construction` includes electrical and plumbing supplies. The user's interest changes according to the actual decoration progress of the condo. Once we have identified that a user is likely to switch from one phase to another, we can begin to recommend the items that belong to the next phase. The ability to incorporate such detailed temporal information is a key improvement over existing recommender system algorithms.

The hidden states include the four phases discussed above, as well as a `NotDecorating` state to indicate that the user is not actively considering items in the Home Decoration. We summarize the user's behavior on the platform on a daily basis. Each day, we assign a strength value to a total of 17 sub-categories (each of the four phases consists of four sub-categories, plus one for no action). The strength value is calculated based on the number of clicks/purchases and serves an indicator of the level of interest of the user in a particular sub-category. The strength value across the 17 sub-categories is normalized to 1. With this formulation, the emission probability is the Multinomial distribution. The duration distribution is Poisson distribution.

The user feature that we provided to the neural network is a 128 dimension embedding trained in a link prediction task using GraphSage. The task is to predict whether a user would click an item solely based on the user embedding and item embedding. Such embedding not only encodes the numerical and categorical information of the users (such as location and income level), but also captures the user-to-item interaction on the platform. We chose 10000 users on the platform that has a large number of clicks/purchases in Home Decoration, and used 180 days of their behavior between January 2018 and July 2018 as training data. The user's behavior between July 1st and July 2nd 2018 is used as validation data.

