# OpenReview forum: "Deep Expectation-Maximization in Hidden Markov Models via Simultaneous Perturbation Stochastic Approximation"
_ICLR.cc/2020/Conference — Reject_

### Official Review · AnonReviewer3 · 2019-10-28
**Official Blind Review #3**

**Rating:** 3

**Review:**

The authors propose to use numerical differentiation (using random perturbation) to approximate the Jacobian of a particular update (essentially equations 5~7) which plays an important role in the estimation of HMMs.  To do so, the authors provide first a concise intro to HMM models (well known stuff in S2), presenting the iteration in detail, jump into their model (cryptically presented in my opinion in S3) and then propose a numerical approximation scheme using SPSA (building upon literature from the 90's, with Theo 1 being the main contribution), before moving onto experiments.

I have found the paper poorly presented. Its general motivation stands on a shaky ground (as illustrated by the choice of words by the authors, see below). In terms of presentation, reminders on HMM are welcome, but unfortunately the authors have not kept the same standard for clarity of notation in Section 3, which makes reading and understanding what the authors are doing quite difficult. Not being a specialist in this field, I have struggled a bit to understand the model itself, and the practical motivation of adding a DNN in the middle of what is otherwise an unrolled back-and-forth between k steps of EM estimation of transition parameters and the addition of a DNN layer. Despite the complexity in the story, what the authors propose is essentially to apply a numerical approximation scheme for Jacobians of these EM updates instead of backprop. Since this is the crux of the contribution,  I feel some more numerical evidence that their approach works compared to baselines (e.g. Hinton et al 2018) is needed. For these reasons my assessment is a bit on the lower side.

- parenthesis bug in b_j(... in Eq.4
- in Eq. 5, index i appears both in numerator (as regular index) and denominator (as sum index)
- what is \Psi in Eq.8 ?
- "While HMM is arguably less prevalent in the era of deep learning": odd way to start an intro. All papers cited date back to more than 2011, 2 in 2006, all the rest in 20th century. This is particularly strange given the few citations to papers >2015 in Section 5.
- the observation sequence o_{t,1:T(u)} is "weakly" indexed by u (since T(u) is just a length)
- What is the \forall u notation below Eq. 9?
- "the number of nodes required to build the forward and backward
probability in the computation graph of an automatic differentiation engine is on the order of O(T^2). Empirically we found this leads to intractable computation cost." since this is critical, where is this empirical evidence? this seems to be a storage problem and cannot be a complexity issue. There are ways to mitigate this problem by only storing partially information, I feel this comparison would add a lot of value to the authors' claim.
- Where is J^{(k)} defined (as opposed to \hat{J}^{(k)}) defined in Eq.14?


**Experience Assessment:**

I do not know much about this area.

**Review Assessment: Checking Correctness Of Derivations And Theory:**

I assessed the sensibility of the derivations and theory.

**Review Assessment: Checking Correctness Of Experiments:**

I did not assess the experiments.

**Review Assessment: Thoroughness In Paper Reading:**

I read the paper at least twice and used my best judgement in assessing the paper.

---

> ### Author Response · Authors · 2019-11-09
> **Thank you for your comments.**
>
> Thank you for your time in reviewing this submission. I would like to further clarify the points you raised.
>
> "parenthesis bug in b_j(... in Eq.4
>  in Eq. 5, index i appears both in numerator (as regular index) and denominator (as sum index)"
> These are indeed typos, we have updated the draft.
>
> "what is \Psi in Eq.8 ?"
> \Psi denotes a mapping, from the observation sequence of a HMM and the parameters of the neural network, to the parameters of the HMM.
>
> "What is the \forall u notation below Eq. 9?"
> $u$ is the index of a particular HMM in a collection of HMMs.  \forall u means we are summing over the loss of all HMMs in the training set.
>
> "Where is J^{(k)} defined (as opposed to \hat{J}^{(k)}) defined in Eq.14?"
> J^{(k)} is the true Jacobian, defined above Eq. 14.
>
> "their model (cryptically presented in my opinion in S3)"
> We basically followed the architecture in Rangapuram et al. In Rangapuram et al, the parameter update is performed by Kalman filter, which only involves a few matrix-to-matrix multiplications and is differentiable. While in this work, the parameter update is via EM iteration. The gradient of the EM iteration is difficult to compute. We provide rigorous analysis to show that the gradient of EM can be reliably approximated.
>
> "some more numerical evidence that their approach works compared to baselines (e.g. Hinton et al 2018) is needed. "
> In my opinion Hinton et al 2018 is concerned with a very different problem and should not be considered as a baseline. We have compared our proposed method to various baselines in Section 5 with satisfactory results.
>
> "since this is critical, where is this empirical evidence? this seems to be a storage problem and cannot be a complexity issue. There are ways to mitigate this problem by only storing partially information"
>
> First of all, this is a storage problem as well as a complexity problem.  If we were to backprop through the EM iteration using an automatic differentiation engine, we need to explicitly represent the forward and backward probability (Equation 2 and 3) in the computation graph. This involves computing as well as storing the value of the forward and backward probability at each step in the observation sequence.
>
> A seemingly plausible way to mitigate this problem is to trade even more computation for less storage, as in Chen et al., 2016. But even that is not possible in this problem. Since we need to explicitly represent the forward and backward probability (Equation 2 and 3) in the computation graph, the \textit{topology} of the computation graph is dependent on the $T$ value. A distinct computation graph has to be built for every distinct $T$ (Section 3, Page 4). This is particularly problematic when the HMMs in a mini-batch comes with different $T$ values, and necessitates gradient approximation methods that do not rely on backpropagation.
>
> Even if $T$ was constant for all samples, we observed that we would hit Tensorflow's 2GB GraphDef limitation with $T = 500$ and batch_size = 256.
>
> "While HMM is arguably less prevalent in the era of deep learning": odd way to start an intro."
> That is a great suggestion. We have updated the draft.
>
> We would like to emphasis that the contribution of this work is by no means limited to HMM related applications. The key contribution of this paper is in incorporating a well-studied classical method (EM algorithm in our case) in a neural network and enable end-to-end training by approximating its gradient. This formulation dramatically expands the choices of network layers and allows network designers to directly apply their domain knowledge in the network architecture.
>
> While there are existing attempts in using domain-specific classical methods in neural networks (for example, Rangapuram et al., 2018 incorporated Kalman filter for time series forecasting. Dong et al., 2018 embedded Lucas-Kanede method for unsupervised facial landmark tracking), they are limited to operations whose gradient are easy to compute. Our key technical contribution (Theorem 1) is in showing that if the gradient of an operation is incomputable, it could potentially be numerically approximated in an efficient and stable manner. It is our hope that this work would enable the community to explore incorporating domain-specific classical method in neural networks, without being limited by the differentiability. We believe this is a fundamental paradigm shift in network architectural design that transcends particular applications.
>
> Reference:
> Chen et al., Training Deep Nets with Sublinear Memory Cost, Arxiv 2016
> Rangapuram et al., Deep state space models for time series forecasting, NIPS 2018
> Dong et al., Supervision-by-Registration: An Unsupervised Approach to Improve the Precision of Facial Landmark Detectors, CVPR 2018

---

### Official Review · AnonReviewer1 · 2019-10-29
**Official Blind Review #1**

**Rating:** 3

**Review:**

The paper is about a method for estimation of parameters of a collection of HMMs and the main contribution is the  combination of classical EM with a neural net.

+ I like the idea of generally approximating gradients in more specific layers that are usually not easy to compute. They clearly formulate their message here and the technical parts of adapting work from prior literature looks solid.
- At the same time I don’t know how much the computational constraints that are formulated are really constraints in practice. A couple of the related works that are cited don’t seem to have these issues. Naively O(T^2) doesn’t really sound like too much of a problem unless the sequences are really long. (In their practical example that isn’t applied to more general recommender systems, this doesn’t really seem to be the case. So it’s unclear )
+ The technical contribution of the gradient estimation seems sound. While I didn’t really go through the proof of convergence, it at least looks rigorous. But I would have to spend a lot more time here to form a well informed opinion.
+ The experiments on synthetic data are clear and further empirically motivate the authors’ work.

The paper is very well written in some parts and in other parts is difficult to understand.

- I am not sure what the benefit of the “e-commerce” application is to the community. The dataset seems to be neither open-source, nor referenced and is insufficiently described. The comparison and conclusion with respect to e.g. GraphSage is hard to interpret as GraphSage is neither explained nor referenced properly (unless I missed it somehow). The authors repeatedly emphasize that their approach works well here but not in the “more general recommender systems scenario”. It would be good if the authors showed something that the rest of the community can directly relate to instead of something that is closed-source and by definition not reproducible.
- I suppose “We apply DENNL in a clearly defined and fast-growing sector on OUR e-Commerce platform, namely Home Decoration” is technically a violation of the blind review if the authors were to now include a reference/link etc. to the dataset. On the other hand, if the dataset remains closed-source then blind review isn’t violated but results aren’t reproducible and hard to follow by the community with the current level of description.

If the authors can comment about the last few points above (especially about open dataset, reproducibility) then I will reconsider raising the rating.

**Experience Assessment:**

I have read many papers in this area.

**Review Assessment: Checking Correctness Of Derivations And Theory:**

I assessed the sensibility of the derivations and theory.

**Review Assessment: Checking Correctness Of Experiments:**

I assessed the sensibility of the experiments.

**Review Assessment: Thoroughness In Paper Reading:**

I read the paper at least twice and used my best judgement in assessing the paper.

---

> ### Author Response · Authors · 2019-11-09
> **Thank you for your comments.**
>
> Thank you for your time in reviewing this submission. I would like to further clarify the points you raised.
>
> "At the same time I don’t know how much the computational constraints that are formulated are really constraints in practice. Naively O(T^2) doesn’t really sound like too much of a problem unless the sequences are really long. "
>
> Note that the length of the observation sequence $T$ varies across different HMM models.  If we were to backprop through the EM iteration using an automatic differentiation engine, we need to explicitly represent the forward and backward probability (Equation 2 and 3) in the computation graph. Thus, the topology of the computation graph is dependent on the $T$ value. Right after we discussed the $O(T^2)$ computation graph size,  we emphasized that since the topology of the computation graph is dependent on the $T$ value, a distinct computation graph has to be built for every distinct $T$ (Section 3, Page 4). This is particularly problematic when the HMMs in a mini-batch comes with different $T$ values, and necessitates gradient approximation methods that do not rely on backpropagation.
>
> Even if $T$ was constant for all samples, we observed that we would hit Tensorflow's 2GB GraphDef limitation with $T = 500$ and batch_size = 256.
>
> "The comparison and conclusion with respect to e.g. GraphSage is hard to interpret as GraphSage is neither explained nor referenced properly (unless I missed it somehow)."
>
> GraphSage (Hamilton et al 2017) was first referenced in Section 1, Page 2. We should have a reference when we first mention GraphSage in the Experiment section as well. Thank you for the comment. We have updated the paper.
>
> "results aren’t reproducible and hard to follow by the community with the current level of description. "
>
> We intend to open source our code and anonymized data upon acceptance of the paper.  Aside from the code and dataset, we consider our theoretical analysis as a major component of this work.
>
> "I am not sure what the benefit of the “e-commerce” application is to the community. "
>
> We readily acknowledge that our e-Commerce application is a niche problem that is of interest to a small audience. However, the idea of incorporating a well-studied classical method (EM algorithm in our case) in a neural network and enable end-to-end training by approximating its gradient is quite novel and could be highly impactful. This formulation dramatically expands the choices of network layers and allows network designers to directly apply their domain knowledge in the network architecture.
>
> While there are existing attempts in using domain-specific classical methods in neural networks (for example, Rangapuram et al., 2018 incorporated Kalman filter for time series forecasting. Dong et al., 2018 embedded Lucas-Kanede method for unsupervised facial landmark tracking), they are limited to operations whose gradient are easy to compute. Our key technical contribution (Theorem 1) is in showing that if the gradient of an operation is incomputable, it could potentially be numerically approximated in an efficient and stable manner. It is our hope that this work would enable the community to explore incorporating domain-specific classical method in neural networks, without being limited by the differentiability. We believe this is a fundamental paradigm shift in network architectural design that transcends particular applications.
>
> Reference:
> Rangapuram et al., Deep state space models for time series forecasting, NIPS 2018
> Dong et al., Supervision-by-Registration: An Unsupervised Approach to Improve the Precision of Facial Landmark Detectors, CVPR 2018

---

### Decision · Program_Chairs · 2019-12-19

**Decision:**

Reject

**Comment:**

The authors propose to use numerical differentiation to approximate the Jacobian while estimating the parameters for a collection of Hidden Markov Models (HMMs). Two reviewers provided detailed and constructive comments, while unanimously rated weak rejection. Reviewer #1 likes the general idea of the work, and consider the contribution to be sound. However, he concerns the reproducibility of the work due to the niche database from e-commerce applications. Reviewer #2 concerns the poor presentation, especially section 3. The authors respond to Reviewers’ concerns but did not change the rating. The ACs concur the concerns and the paper can not be accepted at its current state.